# Are adversities and worries during the COVID-19 pandemic related to sleep quality? Longitudinal analyses of 46,000 UK adults

**Liam Wright** [1,2]*, **Andrew Steptoe**[1], **Daisy Fancourt**[1]

**1** Department of Behavioural Science and Health, University College London, London, United Kingdom,
**2** Department of Epidemiology and Public Health, University College London, London, United Kingdom

* liam.wright.17@ucl.ac.uk

**Data Availability Statement:** The code to produce each of the analyses in this paper is available at https://osf.io/4epqm/. A link to the other data files

## Abstract

### Background

There are concerns that both the experience of adversities during the COVID-19 pandemic and worries about experiencing adversities will have substantial and lasting effects on mental health. One pathway through which both experience of and worries about adversity may impact health is through effects on sleep.

### Methods

We used data from 46,284 UK adults in the COVID-19 Social Study assessed weekly from 01/04/2020-12/05/2020 to study the association between adversities and sleep quality. We studied six categories of adversity including both worries and experiences of: illness with COVID-19, financial difficulty, loss of paid work, difficulties acquiring medication, difficulties accessing food, and threats to personal safety. We used random-effect within-between models to account for all time-invariant confounders.

### Results

Both the total number of adversity experiences and total number of adversity worries were associated with lower quality sleep. Each additional experience was associated with a 1.16 (95% CI = 1.10, 1.22) times higher odds of poor quality sleep while each additional worry was associated with a 1.20 (95% CI = 1.17, 1.22) times higher odds of poor quality sleep. When considering specific experiences and worries, all worries and experiences were significantly related to poorer quality sleep except experiences relating to employment and finances. Having a larger social network offered some buffering effects on associations but there was limited further evidence of moderation by other social or psychiatric factors.

### Conclusion

Poor sleep may be a mechanism by which COVID-19 adversities are affecting mental health. This highlights the importance of interventions that support adaptive coping strategies during the pandemic.

will be added on that website when they become available.

**Funding:** The Covid-19 Social Study was funded by the Nuffield Foundation [WEL/FR-000022583], but the views expressed are those of the authors and not necessarily the Foundation (https://www. nuffieldfoundation.org/). The study was also supported by the MARCH Mental Health Network (https://www.marchnetwork.org/) funded by the Cross-Disciplinary Mental Health Network Plus initiative supported by UK Research and Innovation [ES/S002588/1] (https://www.ukri.org/), and by the Wellcome Trust [221400/Z/20/Z] (https:// wellcome.org/). DF was funded by the Wellcome Trust [205407/Z/16/Z]. LW is funded by the Economic and Social Research Council (https:// esrc.ukri.org/) through the UCL, Bloomsbury and East London Doctoral Training Partnership [ES/ P000592/1] (https://ubel-dtp.ac.uk/). The study was also supported by HealthWise Wales, the Health and Car Research Wales initiative, which is led by Cardiff University in collaboration with SAIL, Swansea University. The funders had no role in study design, data collection and analysis, decision to publish, or preparation of the manuscript. All researchers listed as authors are independent from the funders and all final decisions about the research were taken by the investigators and were unrestricted.

**Competing interests:** The authors have declared that no competing interests exist.

## Introduction

The pandemic of coronavirus disease 2019 (COVID-19) has disrupted lives across the globe. There have been sharp increases in the experience of adversities, both arising from the virus itself (i.e. infection, illness, and possibly death from the disease) and resulting from efforts to contain the disease, such as financial shocks following the loss of employment and income, challenges in accessing food, medication or accommodation, and adverse domestic experiences such as abuse [1–7]. Adversities have followed previous epidemics [8–15], but the scale and length of the COVID-19 pandemic are causing concern that we face manifold public health crises in the years to come [2, 16, 17].

More specifically, there are concerns that adversities experienced during the pandemic will have persisting impacts on physical and mental health [17, 18]. Studies suggest that intimate partner violence [19] and socio-economic adversities such as poverty [20], job loss [21], economic recession [22, 23], and job insecurity [24], can have lasting impacts on mortality and physical and mental health outcomes. Moreover, it is not just the experience of these stressors, but also worries about the potential experience of these stressors, that can affect health, increasing levels of stress and impacting depression and wellbeing [25, 26], as well as affecting physical outcomes such as cardiovascular health [27]. Specifically during COVID-19, we have shown in recent work that there is a relationship between worries and experiences of adversities and poorer mental health [28]. But it is important to understand the biobehavioural pathways through which this relationship exists.

One pathway through which both experience of and worries about adversity may impact health is through effects on sleep [29]. Both worries about adversities and experience of adversities are types of stressors [30]. The stress of experiencing adversities has been shown to impair sleep [31–33], while the stress of worrying about life events has been associated with shorter sleep length and greater sleep disturbance [34, 35]. Numerous biological studies have focused on the pathways underlying these effects, including disruption of HPA axis activity, increased cortisol production, and bidirectional changes between hormonal variation and circadian rhythm [36, 37]. Inadequate sleep may also reinforce the impact of stressors by reducing individual's ability to respond effectively, leading to a maladaptive psychophysiological cycle [38–41]. Impaired sleep is in turn related to worsened physical health outcomes, such as cardiovascular disease, weight gain, and mortality [42, 43], and poorer mental health outcomes, such as anxiety and depression [44]. It is therefore essential to understand whether experience of worries about adversities during the COVID-19 are leading to sleep problems.

While adversity may be related to poorer sleep quality on average, there are several factors that could protect against such effects. First, social support may buffer against stress through the provision of informational or tangible assistance or emotional support [45]. A large body of literature shows that social support is associated with better sleep [46] and with improved physical and mental health outcomes, including lower mortality rates [47]. Further, improved sleep has been identified as a pathway through which social support may affect health [48]. However, decreased face-to-face contact and the increasing prevalence of adversity throughout populations may have reduced the availability and quality of social support during the pandemic [7]. Further, the novel nature of several adversities faced may have reduced the efficacy of informational or tangible assistance aspects of social support. Therefore, an unresolved question is whether social support buffers the association between adversity and sleep quality during lockdown.

A second factor that may be important for the link between adversity and sleep is existing mental health. Studies show that individuals with pre-existing mental health issues may be disproportionately affected psychologically by stressful events. For example, anxiety and

depression can predispose individuals (especially men) to greater stress reactivity [49], while anxiety sensitivity can moderate the relationship between exposure to traumatic events and post-traumatic stress [50]. Further, in previous studies of epidemics, there has been some indication that pre-existing psychiatric conditions are a risk factor for poorer psychological outcomes [8]. However, when considering the link between psychological experiences and sleep, it is possible that individuals with existing mental health conditions may already have poorer sleep, leading to a ceiling effect, such that adversity does not have any further material detrimental effect on sleep [38, 51, 52].

To explore these issues further, the present study used data from a large, longitudinal study of the experiences of adults during the early weeks of the lockdown due to COVID-19 in the UK to explore the time-varying longitudinal relationship between (i) worries about adversity, and (ii) experience of adversity and quality of sleep. Further, it sought to ascertain whether the relationship between adversity and sleep quality was moderated by social support and existing mental health diagnoses.

## Materials and methods

### Participants

We use data from the COVID-19 Social Study; a large panel study of the psychological and social experiences of over 50,000 adults (aged 18+) in the UK during the COVID-19 pandemic. The study commenced on 21 March 2020 and involves online weekly data collection from participants for the duration of the pandemic in the UK. Recruitment into the study is ongoing. The study is not random but does contain a heterogeneous sample. Participants were recruited using three primary approaches. First, snowballing was used, including promoting the study through existing networks and mailing lists (including large databases of adults who had previously consented to be involved in health research across the UK), print and digital media coverage, and social media. Second, more targeted recruitment was undertaken focusing on (i) individuals from a low-income background, (ii) individuals with no or few educational qualifications, and (iii) individuals who were unemployed. Third, the study was promoted via partnerships with third sector organisations to vulnerable groups, including adults with pre-existing mental health conditions, older adults, carers, and people experiencing domestic violence or abuse. The study was approved by the UCL Research Ethics Committee [12467/005] and all participants gave informed consent. The study protocol and user guide (which includes full details on recruitment, retention, data cleaning, weighting and sample demographics) are available at www.covidsocialstudy.org.

Our questions asked about experiences of adversity in the last week, so we focused on data from 1st April 2020 (one week after lockdown commenced) to 12th May 2020, limiting our analysis to participants with two or more waves of data during this period (n = 48,723, observations = 208,057; 80.2% of individuals surveyed between 1 April– 12 May). We used complete case data and excluded participants with complete data in fewer than two data collections (n = 2,439; 5% of eligible participants). This provided a final analytical sample of 46,284 participants (197,372 observations). Note, the sample overlaps that used in our previous work looking at adversities and mental health [28]. Differences in the sample are limited to the small proportion of participants who have missing data on the variables used in this or the previous study.

### Measures

**Adversities.** We study six categories of adversity, each measured weekly (see Table 1). We constructed weekly total adversity worries and total adversity experiences measures by

**Table 1. Questions on adversities.**

| Type of adversity | Adversity worries | Adversity experiences |
|---|---|---|
| COVID-19 illness | Worried about catching COVID-19 | Currently have or previously had suspected or diagnosed COVID-19 |
| Financial difficulty | Worried about finances | Experienced a major cut in household income |
| Loss of paid work | Worried about losing your job/unemployment | Lost one's job or been unable to do paid work |
| Difficulties accessing food | Worried about getting food | Unable to access sufficient food |
| Difficulties acquiring medication | Worried about getting medication | Unable to access required medication |
| Threats to personal safety | Worried about personal safety/security | Experienced being physically harmed or hurt by somebody else or being bullied, controlled, intimidate or psychologically hurt by someone else |

summing the number of adversities present in a given week (range 0–6). For adversity experiences that are likely to be continuing (i.e. once experienced in one week, their effects would likely last into future weeks), we counted them on subsequent waves after they had first occurred. This applied to experiencing suspected/diagnosed COVID-19, loss of paid work, major cut in household income, and abuse victimisation. We considered worries to be one-off events and counted them only in the weeks they were reported.

**Sleep.** Sleep quality was elicited using a single item on sleep: "Over the past week, how has your sleep been?" The item had five response categories: very good, good, average, not good, very poor. Similar single item sleep scales have been shown to be highly correlated with responses to lengthier sleep questionnaires and are widely used in research [53, 54]. To distinguish between minor variations in individual reporting and focus instead on levels of poor sleep quality that are likely to have larger consequences for health, we dichotomised this into a binary variable for not good or poor vs average or better sleep.

**Social support.** We measured social support at first data collection using four separate variables for *loneliness*, *perceived social support*, *social network size*, and *living alone*. Loneliness was measured using the 3-item UCLA-3 loneliness, a short form of the Revised UCLA Loneliness Scale (UCLA-R). Each item is rated with a 3-point rating scale, ranging from "never" to "often", with higher scores indicating greater loneliness. We used the sum score measure (range 3–9).

Perceived social support was measured using an adapted version of the six-item short form of Perceived Social Support Questionnaire (F-SozU K-6). Each item is rated on a 5-point scale from "not true at all" to "very true", with higher scores indicating higher levels of perceived social support. We used the sum score measure (range 6–30). Minor adaptations were made to the language in the scale to make it relevant to experiences during COVID-19 (see S1 Table for a comparison of changes). *Social network size* was measured as number of close friends, with numbers capped at 10+. We included this as a continuous variable.

**Psychiatric illness.** We defined psychiatric illness as reporting a clinically diagnosed mental health problem ("clinically-diagnosed depression", "clinically-diagnosed anxiety", or "another clinically-diagnosed mental health problem") at first data collection.

## Analysis

We used random-effect within-between (REWB) logit models [55] (also known as hybrid models [56]) to explore the association between within-person *change* in adversity experiences and adversity worries and the likelihood of poor quality sleep. Our basic model can be

expressed as follows:

$$P(Bad\ Sleep_{it} = 1) = logit^{-1}(\beta_{0t} + \beta_1 E_{ikt} + \beta_2 \bar{E}_{ik} + \beta_3 W_{ikt} + \beta_4 \bar{W}_{ik} + \beta_L X_t + \alpha_i + \varepsilon_{it})$$

where Bad Sleep$_{it}$ is an indicator for whether individual $i$ reported bad quality sleep at time $t$. $\bar{E}_{ik}$ is the person-specific mean level of adversity experience $k$ across time periods for individual $i$, while $\bar{W}_{ik}$ is the corresponding figure for adversity worries. $E_{ikt}$ and $W_{ikt}$ are the deviations from the person-specific mean values of adversity experiences $k$ and adversity worries $k$ for individual $i$ at time $t$. $X_t$ is a vector of control variables defined below. $\alpha_i$ is the random intercept for individual $i$, which we model as distributed $\sim N(0, \sigma_\alpha^2)$. $\varepsilon_{it}$ is the observation-specific residual error ($\sim N(0, \sigma_\varepsilon^2)$). We ran the models once with "adversity experiences" and "adversity worries" entered separately into the models, so as to ascertain if there was any initial association with sleep, and then re-ran the models with both sets of factors together, to see if results were maintained when mutually adjusting for one another.

Our interest was the sign and size of the coefficients $\beta_1$ and $\beta_3$, which represent the association between within-person change in adversity experiences and adversity worries and the likelihood of poor sleep. We focused on within-person change rather than cross-sectional variation as cross-sectional associations are likely to be confounded by factors such as socio-economic class or personality, which are related to the prevalence of adversity and to sleep. When looking at within-person changes, these characteristics should be fixed, and so associations should not be biased due the influence of these omitted variables on sleep. In fact, in non-linear models such as the logistic model, the coefficients $\beta_1$ and $\beta_3$ are unbiased by time-invariant heterogeneity if the random intercept, $\alpha_i$, is a linear function of the level-2 (between-person) predictors. However, simulations have shown that the extent of bias due to violations of this assumption are limited in practice [55]. Nevertheless, results can still be biased if exposure to new adversities or worries is related to other unobserved changes occurring for the individual.

We estimated several models. In Model 1, we regressed sleep quality on the total number of adversity experiences and total number of adversity worries, both (a) separately and (b) jointly, using the REWB estimator to account for time-invariant heterogeneity across participants. (Variance Inflation Factors suggest multicollinearity was not a problem in this model.) In Model 2, we regressed sleep quality on adversity experiences and adversity worries separately for each category of adversity in turn (finances, personal safety, etc.). In Model 3, we repeated Model 1a including interactions between adversity measures and each social support variable, for each social support variable in turn. In Model 4, we repeated Model 1a including interactions between adversity measures and mental health diagnosis. We adjusted for day of week (categorical) and days since lockdown commenced (continuous) in each regression (person-specific means and deviations from these means). To account for the non-random nature of the sample, all data were weighted to the proportions of gender, age, ethnicity, education and country of living obtained from the Office for National Statistics [57].

We carried out several sensitivity analyses to test the robustness of our results. First, we re-estimated Model 3 using inability to pay bills, rather than major cut in household income, as our measure of experienced financial adversity to differentiate between a change in wealth and a change in wealth that impacts on core financial activity. Second, we repeated each analysis using the sleep item as a continuous variable to test whether results were robust to variable measurement. For these regressions, we used the linear fixed effects estimator which controls for time-invariant confounding by design. Third, we repeated regressions using both the linear probability fixed effect estimator and the fixed effects logit estimators. We did not use the fixed effects logit estimator in the main analysis as the estimator uses information from those whose sleep quality changes only, which may exclude those whose sleep is least responsive to

adversity. Fourth, we repeated our main REWB model for the subset of individuals whose sleep quality changed and compared results against those from the fixed effect logit estimator to assess the possibility of confounding due to time invariant heterogeneity in our main analysis. Analyses were carried out in Stata version 16.0 [58] and R version 3.6.3 [59].

Due to conditions set out by the ethics committee, the data will be made available at the end of 2021. Replication code is available at https://osf.io/4epqm/.

## Results

### Demographics

Descriptive statistics for the exposures and outcomes are shown in Table 2. There was within-variation in each of the measures suggesting REWB was a valid approach. S2 and S3 Tables in the supplementary material displays descriptive statistics for baseline demographic, social support, and mental health diagnosis variables. The weighted sample was 51.3% female, 9.6% from Black and minority ethnic backgrounds, and 19.0% of the sample were aged 18–34, 24.8% aged 35–49, 31.4% aged 50–64 and 24.8% aged 65 and above. Individuals with diagnosed mental illness or with lower social support had worse sleep, on average. S4 Table in the supplementary information displays the sample size by week of data collection. Attrition was less than 10% each week. The average number of follow-ups was 4.26.

Participants reported worry about catching COVID-19 in only 42% of observations overall (S2 Table). We explored the characteristics of those who reported worry about catching COVID-19 at any point during follow-up using bivariate and multivariate logistic models (S1 Fig). When adjusting for all factors simultaneously, older participants, females, individuals with diagnosed physical or mental health conditions, individuals higher in neuroticism, and participants from Northern Ireland were more likely to report worry about catching COVID-19. Conversely, single participants and those with university degrees and high incomes were less likely to report worry about catching COVID-19.

**Table 2. Descriptive statistics.** Weighted figures. Between SD is the standard deviation in participants' average responses. Within SD is the standard deviations in an individual's report, averaged across participants.

| | Variable | Overall Mean | Overall SD | Between SD | Within SD |
|---|---|---|---|---|---|
| | Sleep quality (range 1–5) | 3.12 | 1.08 | 0.95 | 0.51 |
| | Bad Sleep (binary) | 0.29 | 0.45 | 0.37 | 0.26 |
| Experiences | Total number of adversity experiences (range 0–6) | 0.59 | 0.84 | 0.79 | 0.28 |
| | Lost work (binary) | 0.10 | 0.30 | 0.29 | 0.08 |
| | Cut in income (binary) | 0.19 | 0.39 | 0.37 | 0.12 |
| | Unable to access sufficient food (binary) | 0.04 | 0.20 | 0.15 | 0.13 |
| | Unable to access required medication (binary) | 0.03 | 0.16 | 0.12 | 0.11 |
| | Suspected or diagnosed COVID-19 (binary) | 0.13 | 0.34 | 0.33 | 0.08 |
| | Physically or psychologically harmed (binary) | 0.09 | 0.29 | 0.27 | 0.11 |
| Worries | Total number of adversity worries (range 0–6) | 1.30 | 1.32 | 1.15 | 0.66 |
| | Losing job/unemployment (binary) | 0.13 | 0.34 | 0.28 | 0.18 |
| | Finances (binary) | 0.31 | 0.46 | 0.39 | 0.24 |
| | Getting food (binary) | 0.20 | 0.40 | 0.30 | 0.26 |
| | Getting medication (binary) | 0.12 | 0.32 | 0.25 | 0.20 |
| | Catching COVID-19 (binary) | 0.42 | 0.49 | 0.40 | 0.29 |
| | Personal safety (binary) | 0.13 | 0.34 | 0.26 | 0.22 |

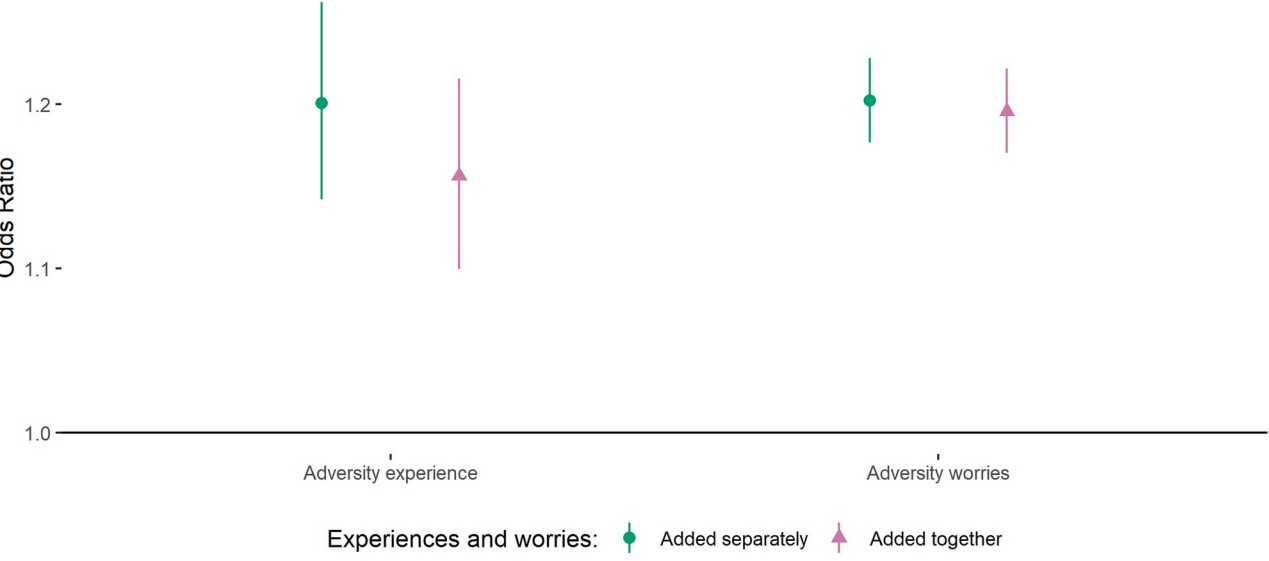

**Fig 1. Associations (with 95% confidence intervals) between (i) change in total number of adversity experiences and (ii) total number of adversity worries and odds of bad quality sleep.** Derived from REWB models. Note: Models either added experiences and worries separately or simultaneously (i.e. so mutually adjusted for one another). Analyses were further adjusted for day of the week and time since lockdown began.

### Associations between adversities and sleep

Both the total number of adversity experiences and total number of adversity worries were associated with lower quality sleep (Fig 1). The inclusion of experiences and worries in the same model slightly reduced the effect size of experiences and had little effect on the effect size of worries. In models including both experiences and worries, each additional experience was associated with a 1.16 (95% CI = 1.10, 1.22) times higher odds of poor quality sleep while each additional worry was associated with a 1.20 (95% CI = 1.17, 1.22) times higher odds of poor quality sleep.

When considering specific experiences and worries, worries were significantly related to poorer quality sleep in every category of adversity (Fig 2). There was some heterogeneity in effect sizes, with the largest effects found for worries about personal safety (OR = 1.43 [1.35, 1.53]), followed by access to medication (OR = 1.39 [1.30, 1.49]), employment (OR = 1.25 [1.16, 1.35]), access to food (OR = 1.24 [1.17, 1.32]), finances (OR = 1.19 [1.12, 1.26]), and catching COVID-19 (OR = 1.18 [1.12, 1.25]).

For experiences, the largest effects were found for access to medication (OR = 1.42 [1.25, 1.61]) and difficulty in accessing food (OR = 1.31 [1.17, 1.47]). Experiencing adversities relating to personal safety such as abuse were also related to poor quality sleep (OR = 1.29 [1.14, 1.47]), as was catching COVID-19 (OR = 1.30 [1.08, 1.54]) (although the confidence intervals were wide potentially indicating heterogeneity in responses). There was some evidence of a relationship between losing work and poor sleep (OR = 1.14 [0.95, 1.38]), but no evidence of a relationship with experiencing a cut in income (OR = 0.95 [0.84, 1.07]).

### Moderators

There was little clear evidence that social support moderated the relationship between sleep quality and adversity experiences (Fig 3; see S5 Table in the supplementary information for interaction term coefficients). For adversity worries (Fig 3), there was evidence that the association between poor quality sleep and adversity worries was *weaker* among those with more

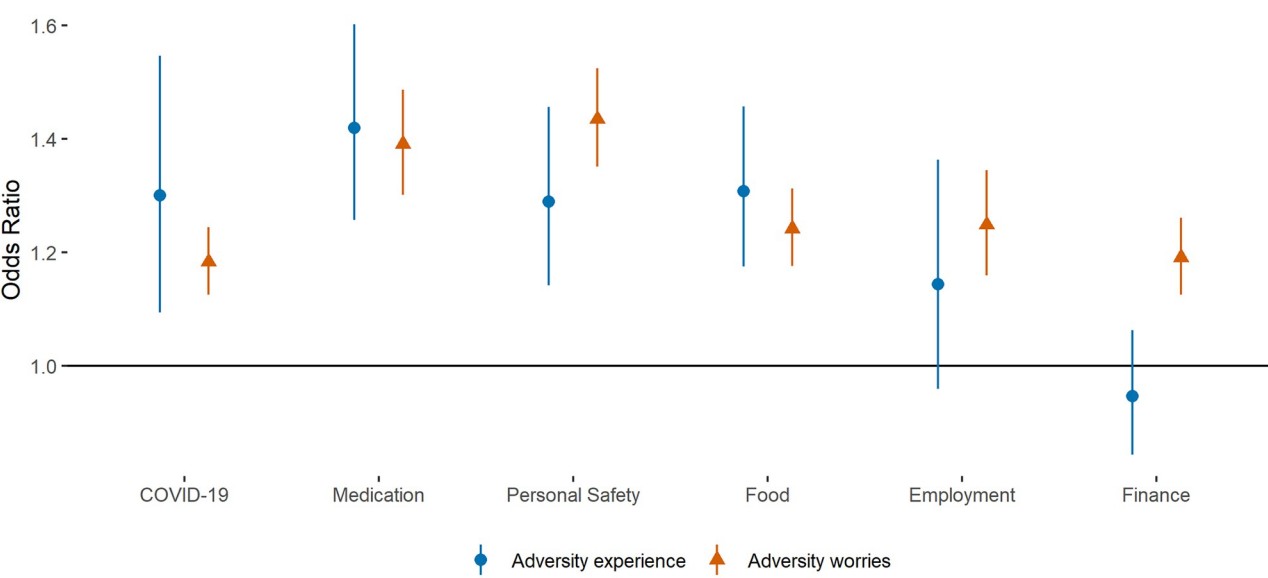

**Fig 2. Associations (with 95% confidence intervals) between (i) change in experience of specific types of adversities or (ii) worries about specific types of adversities and odds of poor sleep.** Derived from REWB models. Experiences and worries were entered into separate models, for each category of adversity in turn. Analyses were further adjusted for day of the week and time since lockdown began.

close friends (OR = 0.97 [0.957, 0.99]). For adversity experiences (Fig 3), there was evidence that the association between poor quality sleep and adversity experiences was *weaker* among those who were lonelier (OR = 0.95 [0.91, 1.00]). For other measures, associations were more tentative (S3 Table).

There was also no evidence of differences in the relationship between worries and sleep quality in people with and without a diagnosed mental illness (Fig 4). There was limited evidence of moderation by mental health for adversity experiences, with larger effects found among those with diagnosed psychiatric conditions (OR = 1.10 [0.99, 1.24]).

## Sensitivity analysis

The results from sensitivity analyses are displayed in the Supplementary Information. Point estimates suggest that inability to pay bills was more highly related to poor sleep quality than reporting a major cut in household income (S2 Fig).

Results using the fixed effects linear probability estimator were qualitatively similar to those from REWB models (S3–S6 Figs). An increase in adversity experiences or adversity worries was association with a ~2% point increase in the probability of poor sleep (S3 Fig). Results using the fixed effects logit estimator, which, as noted above, only uses data from those whose sleep quality changed, were also qualitatively similar to those from REWB models, but produced stronger effect sizes (S7–S10 Figs). An increase in adversity experiences or adversity worries was association with a ~ 4–5% point increase in the probability of poor sleep (S7 Fig). Moderation analyses produced similar effect sizes to those from REWB models, though evidence of moderation according to loneliness was weaker (S9 and S10 Figs and S3 Table). When limiting analyses to individuals whose sleep quality changed, similar results were produced by the REWB and fixed effects logit estimators (S11 Fig), suggesting our main results are not biased due to time invariant heterogeneity.

When analysing sleep quality as a continuous measure, the main findings were qualitatively also similar, with both experiences and worries related to poorer sleep (S12–S15 Figs).

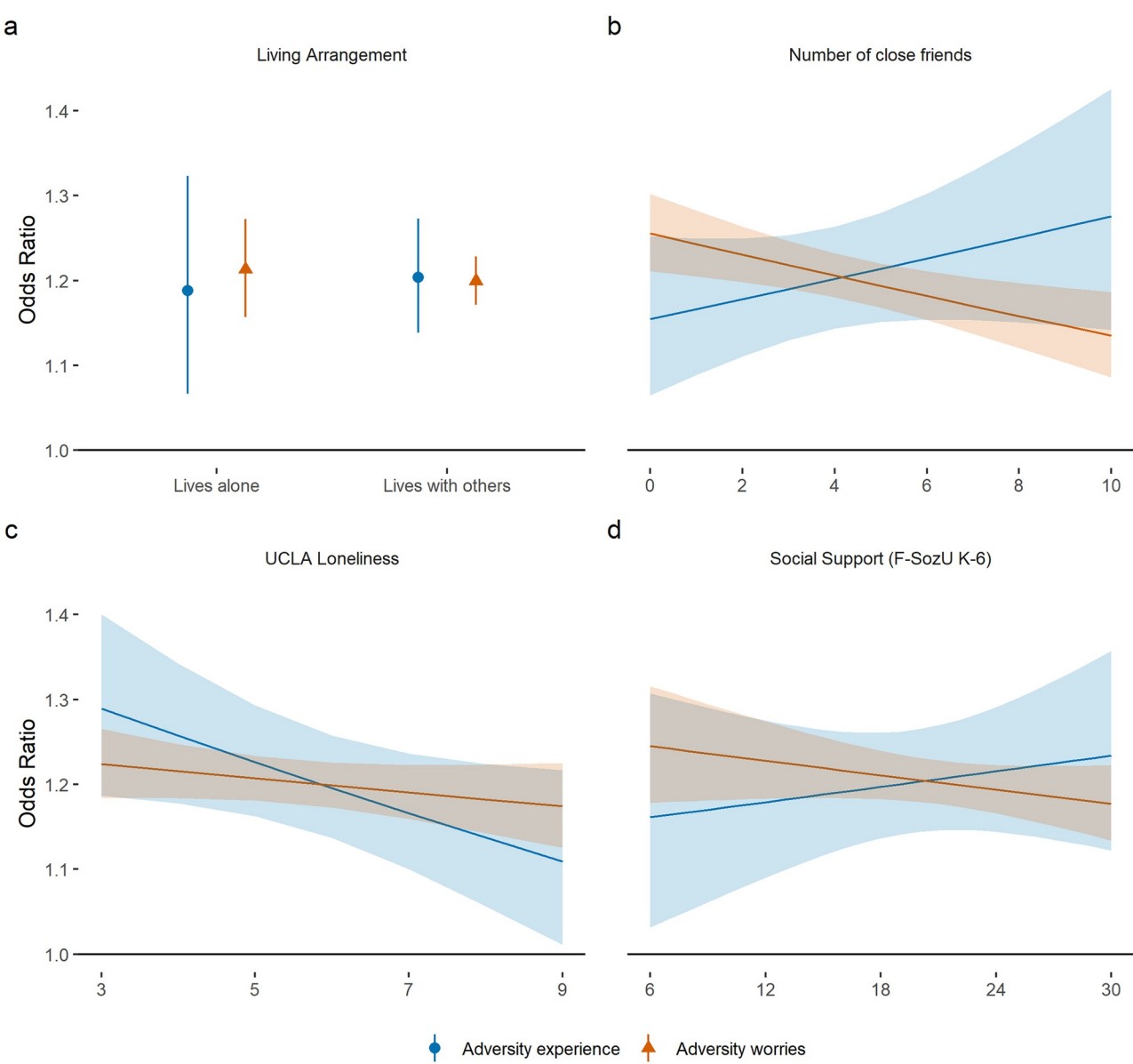

**Fig 3. Associations (with 95% confidence intervals) between (i) change in total number of adversity experiences and (ii) total number of adversity worries and odds of poor quality sleep according to (a) living arrangement, (b) social network size), (c) loneliness, and (d) perceived social support at baseline data collection.** Estimates are from REWB models, with experiences and worries entered into separate models. Analyses were further adjusted for day of the week and time since lockdown began.

However, there was no clear evidence of a moderating role of social support in the association between adversities experiences or worries and sleep (S14 Fig). There was still a moderating role of mental health in the association between adversity experiences and sleep quality (S15 Fig and S3 Table).

One possible issue with our results is our use of a single item sleep quality measure. We compared a similar measure from Wave 4 of the United Kingdom Household Longitudinal Study ("During the past month, how would you rate your sleep quality overall?": 1 = Very good– 4 = Very bad) with other self-report sleep measures collected in that survey (S16 Fig).

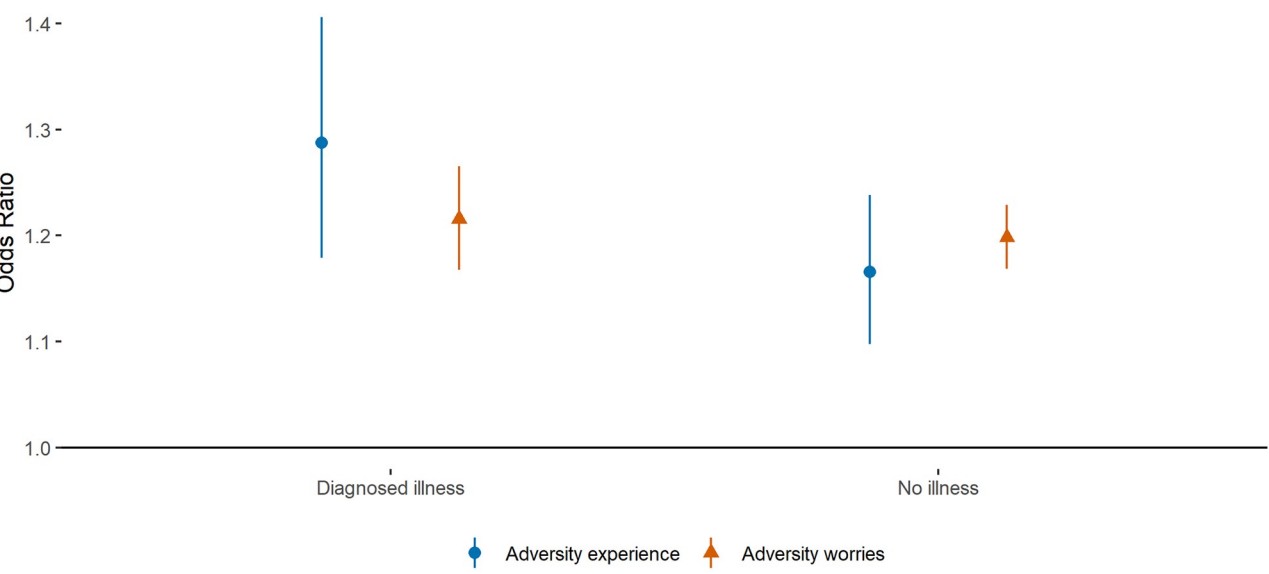

**Fig 4. Associations (with 95% confidence intervals) between (i) change in total number of adversity experiences and (ii) total number of adversity worries and odds of poor quality sleep according to mental health diagnosis at baseline data collection.** Note: Estimates are from REWB models, with experiences and worries entered into separate models. Analyses were further adjusted for day of the week and time since lockdown began.

Lower quality sleep was related to shorter sleep duration, higher wakefulness during the night, greater use of sleep medicines, and greater difficulties getting to sleep.

## Discussion

In this study, we explored the relationship between worries and experience of adversities and quality of sleep during the first lockdown due to COVID-19 in the UK. Cumulative number of worries and experience of adversities were both related to lower quality sleep. When considering specific types of adversities, all types of worries explored were associated with poorer sleep quality, while only specific experiences such as abuse, inabilities to pay bills, access food or medication, and catching COVID-19 showed clear associations with poorer sleep. Effects sizes were small: additional adversity experience or worries were related to approximately a 2%-point higher likelihood of poor quality sleep, on average. Having more close friends helped to moderate the relationship between worries and sleep but there was weaker evidence that other social factors had any clear protective buffering effects.

This study supports findings from emerging research on COVID-19, which has suggested that sleep is being adversely affected amongst people during the pandemic [60]. The clear relationship between both specific and cumulative worries and poor sleep echoes findings about the adverse effects of stress on sleep from a number of previous studies [31–33]. However, it is notable that only specific experiences were related to poor sleep. These related specifically to difficulties in accessing food and medication, experience of abuse, and contracting COVID-19. In particular, experience of domestic violence has previously been well-researched in relation to sleep, with studies notably suggesting that fear of future abuse and nightmares can disrupt sleep [61]. There has also been increasing research focus on the neuropsychiatric effects of coronavirus infections, with suggestions that sleep disturbance can follow from infection [62]. This could explain the findings showing a relationship between having COVID-19 and impaired sleep. Notably, we didn't find a clear relationship between experiencing loss of work or cuts in household income and impaired sleep, although worry about these things was

associated with poorer sleep. It is possible that consequences may take time to arise. For instance, loss of paid work or cuts in income may impact sleep only following repeated rejections during job search or when reduced incomes begin to impact living standards [63, 64]. Financial adversities may also have been anticipated such that effects were felt in anticipation of the financial adversities, and high strain work may itself have adversely impacted sleep [33]. The effect of job loss on stress may also have been counterbalanced by increased leisure time [65].

Our results also found only limited evidence of buffering of these associations by social factors, in perhaps contradictory direction. Having more close friends appeared to buffer the association between adversity worries and sleep. This is notable given that social contact with friends was not permitted during the period follow due to lockdowns. It could suggest that having a circle of friends provides reassurance even if their support is not explicitly drawn on. But it is notable that other similar factors, such as social support, did not act as moderators. Cross-sectional studies during the pandemic have founds associations between social support and better sleep quality [66, 67]. However, these studies did not look at the relationship between social support and adversity worries or experiences, suggesting that while social support may be linked with sleep, it may not go so far as to moderate the relationship between sleep and other factors. There has been some previous research finding that social support can moderate the relationship between occupational stress and sleep outside of pandemic settings [68]. But social support at work (available on a daily basis alongside one's occupational stressors) may play a more meaningful role than during a pandemic when one may know one has social support available if needed but may not be necessarily drawing on that support, thereby rendering it a hypothetical rather than realised type of social support.

There was also some evidence that loneliness buffered the association with adversity experiences. However, this was non-robust when testing it in sensitivity analyses, so remains to be explored further. For other social factors there was only limited evidence of any moderating effect. It is possible that decreased social interaction or limited face-to-face contact with social networks may have reduced any protective effects [7].

Further, it is interesting that there was only limited evidence of moderation by mental illness. Anxiety and depression can predispose individuals to greater stress reactivity [49], and our results suggested there could be slightly larger effects amongst those diagnosed psychiatric conditions. But results were not clear, and both those with and without psychiatric conditions are at risk of poor sleep as a result of adversities. This echoes other research showing how adversities and stresses affect not just those at high risk but also the broader population [7].

This study has a number of strengths. We used data from a large, heterogeneous sample, which was weighted to UK population proportions according to major socio-demographic characteristics. The data covered six weeks of lockdown in the UK, providing rich longitudinal data that allowed us to estimate the relationship between adversity and change in sleep *within* individuals, rather than rely on cross-sectional variation, which would likely be confounded by time-invariant heterogeneity across individuals.

The study has several limitations, however. While we were about to account for confounding through time-invariant heterogeneity, time-varying factors that remain unobserved may explain results. Further, whilst is appears logical that poor sleep itself cannot cause adverse experiences, there is likely a bidirectional relationship between worries and poor sleep, and worries may pre-date experiences. Nevertheless, our analyses suggest that both worries and experiences are independently associated with poor sleep. Another limitation was our use of a single item five-category self-report measure of sleep quality. This did not provide detail on which aspect of sleep was most affected (e.g. duration, onset, interruption etc) and may have

lacked sufficient variation and validity to accurately estimate effects. Indeed, self-reports of sleep have been found to be worse in psychiatric patients, which may also have biased responses [69]. It is possible that individuals experiencing worries or adversities may have perceived their sleep to be worse, but without substantial variation in the core qualitative parameters of sleep. However, single item sleep scales have been shown to be highly correlated with responses to lengthier sleep questionnaires and are widely used in research [53]. Further, we also show that a similar item collected in a representative UK household panel survey is related to aspects of sleep such as sleep duration and use of sleep medicines.

The sampling in the COVID-19 Social Study was not random. While we deliberately sampled from groups such as individuals of low socio-economic position and individuals with existing mental illness, individuals with particularly extreme experiences may not have been adequately captured in the study. Further, individuals experiencing extreme adversity during the lockdown may have been more likely to withdraw from the study. While our statistical approach allowed for an unbalanced panel, non-random attrition could have biased results. Social support was measured at first data collection, which for many was after lockdown began. Responses to these questions could have been affected by adversities experienced already. We also focused on just six types of adversities, but other types of adversity were not included in the study, notably those relating to interpersonal relationships and bereavement. Finally, our study only followed individuals up over a period of weeks. It remains for future studies to assess how experience of adversities during the COVID-19 pandemic relates to sleep–and to health–in the long-term.

Previous studies have shown that experience and worries about adversities during COVID-19 are associated with poorer mental health. The results presented here suggest that poor sleep may be a mechanism by which such adversities are affecting mental health. Worries about adversities were related to poorer quality sleep over time, as was cumulative load of adverse experiences was also associated with poorer quality sleep. But only specific adversities such as those relating to personal safety, catching COVID-19, or challenges in accessing food and medication showed clear associations with poor sleep on their own. These results were relatively consistent amongst those with and without a diagnosed mental illness. Having a larger social network had some protective effects, but other social factors had more limited moderating effects on the relationship. These results may be generalisable to non-pandemic settings, showing how two different types of stressors (experiences and worries) are similarly related to sleep. Further, many of the measures of stressors we focused on (including both the worries about and experiences of adversities) can be experienced in daily life. However, the results also have an immediate relevance to supporting individuals during the current pandemic. They suggest the importance of interventions that seek to reassure individuals and enable adaptive coping strategies. Given the challenges in providing face-to-face mental health support to individuals during lockdowns due to COVID-19, these findings highlight the importance of developing online and remote interventions that could provide such support, both as COVID-19 continues and in preparation for future pandemics.

## Supporting information

**S1 Table. Comparison of items in the original and revised perceived social support questionnaire (F-SozU K-6).**
(DOCX)

**S2 Table. Descriptive statistics, social support and psychiatric diagnosis.** Weighted figures.
(DOCX)

**S3 Table. Sample descriptive statistics.**
(DOCX)

**S4 Table. Sample size by week.**
(DOCX)

**S5 Table. Estimated interaction effects between social support and diagnosed psychiatric illness and deviation in total adversity experiences and adversity worries.**
(DOCX)

**S1 Fig. Association between demographic characteristics and reporting worry about catching COVID-19 during any data collection.** Derived from bivariate and multivariate survey weighted logistic regression models. Multivariate models include adjustment for each factor simultaneously. Big-Five personality traits are scaled (mean = 0, SD = 1).
(TIF)

**S2 Fig. Associations between (i) cut in income and (ii) inability to pay bills and odds of bad quality sleep.** Derived from REWB models.
(TIF)

**S3 Fig. Associations between (i) change in total number of adversity experiences and (ii) total number of adversity worries and probability of bad quality sleep.** Derived from linear probability fixed effects models.
(TIF)

**S4 Fig. Associations between (i) change in experience of specific types of adversities or (ii) worries about specific types of adversities and probability of poor sleep.** Derived from linear probability fixed effects models.
(TIF)

**S5 Fig. Associations between (i) change in total number of adversity experiences and (ii) total number of adversity worries and probability of poor quality sleep according to (a) living arrangement, (b) social network size), (c) loneliness, and (d) perceived social support at baseline data collection.** Derived from linear probability fixed effects models.
(TIF)

**S6 Fig. Associations between (i) change in total number of adversity experiences and (ii) total number of adversity worries and probability of poor quality sleep according to mental health diagnosis at baseline data collection.** Derived from linear probability fixed effects models.
(TIF)

**S7 Fig. Associations between (i) change in total number of adversity experiences and (ii) total number of adversity worries and probability of bad quality sleep.** Derived from fixed effects logit models.
(TIF)

**S8 Fig. Associations between (i) change in experience of specific types of adversities or (ii) worries about specific types of adversities and probability of poor sleep.** Derived from fixed effects logit models.
(TIF)

**S9 Fig. Associations between (i) change in total number of adversity experiences and (ii) total number of adversity worries and probability of poor quality sleep according to (a)**

living arrangement, (b) social network size), (c) loneliness, and (d) perceived social support at baseline data collection. Derived from fixed effects logit models.
(TIF)

**S10 Fig. Associations between (i) change in total number of adversity experiences and (ii) total number of adversity worries and probability of poor quality sleep according to mental health diagnosis at baseline data collection.** Derived from fixed effects logit models.
(TIF)

**S11 Fig. Associations between (i) change in experience of all/specific types of adversities or (ii) worries about all/specific types of adversities and probability of poor sleep.** Derived from fixed effects logit models and REWB models, where sample is those whose sleep quality changed during follow-up period.
(TIF)

**S12 Fig. Associations between (i) change in total number of adversity experiences and (ii) total number of adversity worries and (continuous) sleep quality.** Derived from fixed effects models.
(TIF)

**S13 Fig. Associations between (i) change in experience of specific types of adversities or (ii) worries about specific types of adversities and (continuous) sleep quality.** Derived from fixed effects models.
(TIF)

**S14 Fig. Associations between (i) change in total number of adversity experiences and (ii) total number of adversity worries and (continuous) sleep quality according to (a) living arrangement, (b) social network size), (c) loneliness, and (d) perceived social support at baseline data collection.** Derived from fixed effects models.
(TIF)

**S15 Fig. Associations between (i) change in total number of adversity experiences and (ii) total number of adversity worries and (continuous) quality sleep quality according to mental health diagnosis at baseline data collection.** Derived from fixed effects models.
(TIF)

**S16 Fig. Convergent validity of a single item sleep quality measure from Wave 4 of the United Kingdom household longitudinal study with other measures of sleep from the same survey.** Sleep quality: "During the past month, how would you rate your sleep quality overall?". Sleep hours: "How many hours of actual sleep did you usually get per night during the last month?". 30+ minutes to sleep: "During the past month, how often have you had trouble sleeping because you. . . cannot get to sleep within 30 minutes?". Waking frequency: "(During the past month, how often have you had trouble sleeping because you. . .) wake up in the middle of the night or early in the morning?". Sleep medicine: "During the past month, how often have you taken medicine (prescribed or "over the counter") to help you sleep?". The correlation between sleep quality and sleep time is 0.51.
(TIF)

**S1 File. STROBE checklist for observational studies.**
(DOCX)

## Acknowledgments

The researchers are grateful for the support of a number of organisations with their recruitment efforts including: the UKRI Mental Health Networks, Find Out Now, UCL BioResource, SEO Works, FieldworkHub, and Optimal Workshop.

## Patient and public involvement

The research questions in the UCL COVID-19 Social Study built on patient and public involvement as part of the UKRI MARCH Mental Health Research Network, which focuses on social, cultural and community engagement and mental health. This highlighted priority research questions and measures for this study. Patients and the public were additionally involved in the recruitment of participants to the study and are actively involved in plans for the dissemination of findings from the study.

## Author Contributions

**Conceptualization:** Liam Wright, Andrew Steptoe, Daisy Fancourt.

**Data curation:** Liam Wright.

**Formal analysis:** Liam Wright.

**Funding acquisition:** Andrew Steptoe, Daisy Fancourt.

**Methodology:** Liam Wright, Daisy Fancourt.

**Supervision:** Andrew Steptoe, Daisy Fancourt.

**Validation:** Daisy Fancourt.

**Visualization:** Liam Wright.

**Writing – original draft:** Daisy Fancourt.

**Writing – review & editing:** Liam Wright, Andrew Steptoe.

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
