## [Decision Letter · Decision Letter 0]

21 Dec 2020

PONE-D-20-32170

Are adversities and worries during the COVID-19 pandemic related to sleep quality? Longitudinal analyses of 46,000 UK adults

PLOS ONE

Dear Dr. Wright,

Thank you for submitting your manuscript to PLOS ONE. After careful consideration, we feel that it has merit but does not fully meet PLOS ONE’s publication criteria as it currently stands. Therefore, we invite you to submit a revised version of the manuscript that addresses the points raised during the review process.

Please submit your revised manuscript by February 20, 2021. If you will need more time than this to complete your revisions, please reply to this message or contact the journal office at plosone@plos.org. Please include the following items when submitting your revised manuscript:

We look forward to receiving your revised manuscript.

Kind regards,

Kristin Vickers, Ph.D.

Academic Editor

PLOS ONE

Journal Requirements:

2. Please provide the following information in your Methods section relating to the inter-views conducted in the study. In this case, please detail 1) the training and background of the inter-viewer, 2) where the interviews took place, 3) how long each interview lasted.

3.We note that you have indicated that data from this study are available upon request. PLOS only allows data to be available upon request if there are legal or ethical restrictions on sharing data publicly. For information on unacceptable data access restrictions, please see http://journals.plos.org/plosone/s/data-availability#loc-unacceptable-data-access-restrictions.

4. We note you have included a table to which you do not refer in the text of your manuscript. Please ensure that you refer to Table 2 in your text; if accepted, production will need this reference to link the reader to the Table.

5. We noticed you have some minor occurrence of overlapping text with the following preprint, of which you are an author, which needs to be addressed:

- https://www.medrxiv.org/content/10.1101/2020.05.14.20101717v2

In your revision ensure you cite all your sources (including your own works), and quote or rephrase any duplicated text outside the methods section. Further consideration is dependent on these concerns being addressed.

Additional Editor Comments (if provided):

Dear Dr. Wright,

Thank you for submitting your interesting research to PLOS One. I apologize for the delay in our response; several reviewers who agreed initially were not able to complete their reviews. In order to get a decision to you as soon as possible, we are proceeding with two reviews, one from a content expert and one from a statistical expert. The content expert (reviewer one) recommends a major revision, and I agree with this decision. In particular, I ask you to consider whether more substantive content might be added to the Discussion section (reviewer one). Both reviewers provide other thoughtful and important feedback, and I will not repeat the comments here, but ask you to action these (or if you choose not to action a particular suggestion, please explain why). Of especial importance, both reviewers note that there is a paper that may have an overlapping sample (meaning that these manuscripts are related). PLOS ONE has guidelines when different publications are related. If there is no overlap between this manuscript and the other publication, please let me know that. If there is any overlap, please follow the PLOS ONE guidelines at https://journals.plos.org/plosone/s/submission-guidelines#loc-related-manuscripts. Specifically, upon your submission of your revised manuscript, please indicate whether there are any related manuscripts under consideration or published elsewhere. If related work has been submitted or published elsewhere, please include a copy of it with your revised manuscript and describe its relation to the submitted work. I will also ask that that the authors make it clear in the revised manuscript that results from a related dataset have been published previously or are under consideration for publication, if applicable.

Reviewers' comments:

Reviewer's Responses to Questions

**Comments to the Author**

1. Is the manuscript technically sound, and do the data support the conclusions?

Reviewer #1: Yes

Reviewer #2: Yes

2. Has the statistical analysis been performed appropriately and rigorously? 

Reviewer #1: I Don't Know

Reviewer #2: Yes

3. Have the authors made all data underlying the findings in their manuscript fully available?

Reviewer #1: No

Reviewer #2: Yes

4. Is the manuscript presented in an intelligible fashion and written in standard English?

Reviewer #1: Yes

Reviewer #2: Yes

5. Review Comments to the Author

Reviewer #1: This manuscript examines how adverse events and worries affect sleep in these days of the COVID pandemic. The paper is obviously timely, and the large stratified sample (~46,000) from the UK is a strength of the study. Both worries and adverse events had a small deleterious effect on sleep, with worries being slightly more impactful. Interestingly, social network had relatively little impact on this effect.

The paper is heavy on statistical analysis and light on discussion. A different reviewer would be better to comment on that statistical analysis.

The authors have an overlapping paper (their reference number 28) which used the same sample in an effort to understand the impact of adversities and worries on mental health. Slicing their analyses so finely, makes it difficult to examine the interplay of sleep and mental health.

There is a typo on Table 1. The rows “difficulties acquiring medication” and “difficulties accessing food” are reversed.

There is a glitch in the manuscript citation in line 1 of the demographics results section.

It is intriguing that so few people were worried about “catching COVID-19” (see table following line 210). That sample would be interesting in its own right. What sorts of life situations contribute to such worries or conversely make some people oblivious to the worries?

I am not sure that the figures add that much value to the paper.

Reviewer #2: Important note: This review pertains only to ‘statistical aspects’ of the study and so ‘clinical aspects’ [like medical importance, relevance of the study, ‘clinical significance and implication(s)’ of the whole study, etc.] are to be evaluated [should be assessed] separately/independently. Further please note that any ‘statistical review’ is generally done under the assumption that (such) study specific methodological [as well as execution] issues are perfectly taken care of by the investigator(s). This review is not an exception to that and so does not cover clinical aspects {however, seldom comments are made only if those issues are intimately / scientifically related & intermingle with ‘statistical aspects’ of the study}. Agreed that ‘statistical methods’ are used as just tools here, however, they are vital part of methodology [and so should be given due importance].

COMMENTS: Your ABSTRACT is well drafted but assay type. Please note that it is preferable [refer to item 1b of CONSORT checklist 2010: Structured summary of trial design, methods, results, and conclusions] to divide the ABSTRACT with small sections like ‘Objective(s)’, ‘Methods’, ‘Results’, ‘Conclusions’, etc. which is an accepted practice of most good/standard journals [including PLOS-ONE]. It will definitely be more informative then, I guess, whatever the article type may be.

I find lot many things common in this article with that of reference number 28 [Wright L, Steptoe A, Fancourt D. How are adversities during COVID-19 affecting mental health? Differential associations for worries and experiences and implications for policy. medRxiv. 2020 May 19;2020.05.14.20101717] given here (though the focus now is on pathway through effects on quality of sleep). It is natural as the data-base is probably same and the authors are same (in same sequence), however (though that is alright), some comments on or summary of overlap is desirable to appear in this article in brief. Although the topic [Are adversities and worries during the COVID-19 pandemic related to sleep quality?] dealt in this article is interesting, at least few things, if not many, here are ‘obvious’ in my opinion.

What exactly you mean by “The study is not random but does contain a well-stratified sample” (lines 93-4)? [no stratified sampling done then is this conclusion/finding/observation based on post-stratification?] Rest of the methodology and statistical methods used are correct [very good]. It is appreciated that ‘logit’ is used in hybrid models {though ‘logit’ is the only appropriate choice, often mistakenly ‘linear’ model(s) are used}.

Nothing more could be suggested/highlighted. Minor revision is recommended.

6. PLOS authors have the option to publish the peer review history of their article (what does this mean?). If published, this will include your full peer review and any attached files.

Reviewer #1: No

Reviewer #2: No

---

## [Author Response · Author response to Decision Letter 0]

20 Jan 2021

Dear Dr Vickers,

We hope you had a restful Christmas break.

Thank you to you and the reviewers for accepting our work for publication with revisions. We have uploaded an amended manuscript based on your comments. In the attached file we respond to each of the points raised directly.

Best wishes,

Daisy

---

## [Decision Letter · Decision Letter 1]

18 Feb 2021

PONE-D-20-32170R1

Are adversities and worries during the COVID-19 pandemic related to sleep quality? Longitudinal analyses of 46,000 UK adults

PLOS ONE

Dear Dr. Wright,

Thank you for submitting your revised manuscript to PLOS ONE. After careful consideration, we feel that it has merit but does not fully meet PLOS ONE’s publication criteria as it currently stands. Therefore, we invite you to submit a revised version of the manuscript that addresses the points raised during the review process.

The decision on this revision is Minor Revision. Please address the comment of the reviewer. Also, please consider whether you can expand on how this manuscript is related to the pre-print (now accepted for publication; congratulations). I see one sentence that references your prior work (reference 28). “Specifically during COVID-19, we have shown in recent work that there is a relationship between worries and experiences of adversities and poorer mental health [28].”  Please consider if there is other information you could add to clarify how these studies are related (was the same sample used, for example; were the 41,909 UK adults in the pre-print also included in the 46,284 participants in this work). I look forward to receiving your minor revision of this manuscript.   

We look forward to receiving your revised manuscript.

Kind regards,

Kristin Vickers, Ph.D.

Academic Editor

PLOS ONE

Reviewers' comments:

Reviewer's Responses to Questions

**Comments to the Author**

1. If the authors have adequately addressed your comments raised in a previous round of review and you feel that this manuscript is now acceptable for publication, you may indicate that here to bypass the “Comments to the Author” section, enter your conflict of interest statement in the “Confidential to Editor” section, and submit your "Accept" recommendation.

Reviewer #2: (No Response)

2. Is the manuscript technically sound, and do the data support the conclusions?

Reviewer #2: (No Response)

3. Has the statistical analysis been performed appropriately and rigorously? 

Reviewer #2: (No Response)

4. Have the authors made all data underlying the findings in their manuscript fully available?

Reviewer #2: (No Response)

5. Is the manuscript presented in an intelligible fashion and written in standard English?

Reviewer #2: (No Response)

6. Review Comments to the Author

Reviewer #2: COMMENTS: Please note that the meaning of the terms “well stratified” and “heterogeneous” are not same Therefore, your response/answer to my comment/question regarding stratification that “We meant that the sample was heterogeneous in the demographics represented” is not acceptable {and even liked} as what you meant cannot be understood [how do expect that?] automatically. Remember that this is a scientific/academic document and so all details should be accurately/clearly/correctly communicated, and terms used carefully. Agreed that English is not my mother tongue however, I never knew that the term “well stratified” is alternative to “heterogeneous”.

It is well known that the evidence from secondary data analyses of survey data is/are considered as very low level research evidence, however, importance of study on ‘COVID-19 adversities and sleep quality’ is of enough value and so this view was not taken by this reviewer. I guess, what I said on earlier draft [that “Nothing more could be suggested/highlighted” after making just two very minor comments] is a clear indication. Does not that so?

7. PLOS authors have the option to publish the peer review history of their article (what does this mean?). If published, this will include your full peer review and any attached files.

Reviewer #2: No

---

## [Author Response · Author response to Decision Letter 1]

18 Feb 2021

Dear Dr Vickers,

Please see the attached files for our response to your and Reviewer #2's comments.

With best wishes,

Liam

---

## [Decision Letter · Decision Letter 2]

9 Mar 2021

Are adversities and worries during the COVID-19 pandemic related to sleep quality? Longitudinal analyses of 46,000 UK adults

PONE-D-20-32170R2

Dear Dr. Wright,

We’re pleased to inform you that your manuscript has been judged scientifically suitable for publication and will be formally accepted for publication once it meets all outstanding technical requirements.

Kind regards,

Kristin Vickers, Ph.D.

Academic Editor

PLOS ONE

Reviewers' comments:

Reviewer's Responses to Questions

**Comments to the Author**

1. If the authors have adequately addressed your comments raised in a previous round of review and you feel that this manuscript is now acceptable for publication, you may indicate that here to bypass the “Comments to the Author” section, enter your conflict of interest statement in the “Confidential to Editor” section, and submit your "Accept" recommendation.

Reviewer #2: (No Response)

2. Is the manuscript technically sound, and do the data support the conclusions?

Reviewer #2: (No Response)

3. Has the statistical analysis been performed appropriately and rigorously? 

Reviewer #2: (No Response)

4. Have the authors made all data underlying the findings in their manuscript fully available?

Reviewer #2: (No Response)

5. Is the manuscript presented in an intelligible fashion and written in standard English?

Reviewer #2: (No Response)

6. Review Comments to the Author

Reviewer #2: COMMENTS: All the comments made on earlier draft(s) by me (and hopefully by other respected reviewers also) were/are attended positively/adequately, I had already expressed satisfaction and the manuscript is improved a lot. I recommend acceptance.

7. PLOS authors have the option to publish the peer review history of their article (what does this mean?). If published, this will include your full peer review and any attached files.

Reviewer #2: **Yes: **Dr. Sanjeev Sarmukaddam

---

## [Editor Report · Acceptance letter]

15 Mar 2021

PONE-D-20-32170R2 

Are adversities and worries during the COVID-19 pandemic related to sleep quality? Longitudinal analyses of 46,000 UK adults 

Dear Dr. Wright:

I'm pleased to inform you that your manuscript has been deemed suitable for publication in PLOS ONE. Congratulations! Your manuscript is now with our production department. 

Kind regards, 

on behalf of

Dr. Kristin Vickers 

Academic Editor

PLOS ONE